# Mapping the Distributions of Mosquitoes and Mosquito-Borne Arboviruses in China

**DOI:** 10.3390/v14040691

**Published:** 2022-03-27

**Authors:** Tao Wang, Zheng-Wei Fan, Yang Ji, Jin-Jin Chen, Guo-Ping Zhao, Wen-Hui Zhang, Hai-Yang Zhang, Bao-Gui Jiang, Qiang Xu, Chen-Long Lv, Xiao-Ai Zhang, Hao Li, Yang Yang, Li-Qun Fang, Wei Liu

**Affiliations:** 1State Key Laboratory of Pathogen and Biosecurity, Beijing Institute of Microbiology and Epidemiology, Beijing 100071, China; wang_tao0515@126.com (T.W.); zwfan9128@163.com (Z.-W.F.); loveyangguang787@foxmail.com (Y.J.); chenyever@yeah.net (J.-J.C.); 15122056576@163.com (G.-P.Z.); sxsdtzwh@163.com (W.-H.Z.); zhhy_ql@163.com (H.-Y.Z.); jiangbaogui@hotmail.com (B.-G.J.); xq854821707@163.com (Q.X.); tjjys2012@163.com (C.-L.L.); babylovehopi@163.com (X.-A.Z.); 2College of Public Health and Health Professions and Emerging Pathogens Institute, University of Florida, Gainesville, FL 32611, USA

**Keywords:** mosquitoes, arboviruses, mosquito-borne diseases, distribution, risk determinants, China

## Abstract

The geographic expansion of mosquitos is associated with a rising frequency of outbreaks of mosquito-borne diseases (MBD) worldwide. We collected occurrence locations and times of mosquito species, mosquito-borne arboviruses, and MBDs in the mainland of China in 1954−2020. We mapped the spatial distributions of mosquitoes and arboviruses at the county level, and we used machine learning algorithms to assess contributions of ecoclimatic, socioenvironmental, and biological factors to the spatial distributions of 26 predominant mosquito species and two MBDs associated with high disease burden. Altogether, 339 mosquito species and 35 arboviruses were mapped at the county level. *Culex tritaeniorhynchus* is found to harbor the highest variety of arboviruses (19 species), followed by *Anopheles sinensis* (11) and *Culex pipiens quinquefasciatus* (9). Temperature seasonality, annual precipitation, and mammalian richness were the three most important contributors to the spatial distributions of most of the 26 predominant mosquito species. The model-predicted suitable habitats are 60–664% larger in size than what have been observed, indicating the possibility of severe under-detection. The spatial distribution of major mosquito species in China is likely to be under-estimated by current field observations. More active surveillance is needed to investigate the mosquito species in specific areas where investigation is missing but model-predicted probability is high.

## 1. Introduction

Mosquitoes, a family of blood-sucking arthropods, are considered to be one of the most influential vectors for harboring and transmitting a variety of mosquito-borne diseases (MBD). MBDs have caused high disease burden among humans, including but not limited to dengue fever, chikungunya fever, Zika virus disease, yellow fever, Japanese encephalitis (JE), and malaria, with around 96 million, 693 thousand, 500 thousand, 200 thousand, 68 thousand, and 212 million cases per year, respectively [1,2,3,4]. Except for African swine fever virus, a DNA arbovirus transmitted by ticks, the majority of arboviruses are RNA viruses [5,6]. Compared with DNA viruses, RNA viruses exhibit greater genetic plasticity and higher mutation rates, which allow them to accommodate a cycle of alternating replication in disparate vertebrate and invertebrate hosts [7,8]. This means that RNA viruses are likely to expand the variety of host animals and vectors, as well as the scope of the spread. A typical example is chikungunya virus (CHIKV), which was native to Africa and was transmitted by *Aedes* (*Ae.*) *aegypti* at first, but has spread to other continents since 2005, likely driven by a mutation increasing its capacity to infect, reproduce in, and be transmitted by *Ae. albopictus* [9]. Moreover, there is a diversity of serotypes and genotypes for arboviruses, such as the dengue virus (DENV). There are four notifiable MBDs in the Chinese notifiable infectious disease system, including filariasis, malaria, dengue fever, and JE. Among them, filariasis and malaria were caused by two parasites, i.e., filarial worm and *Plasmodium*, and were certified as eliminated by the World Health Organization (WHO) in 2007 and 2021, respectively [10,11]. Meanwhile, dengue fever and JE had high disease burdens of MBDs, with annual incidence rates of 1.08 and 0.08 per 100,000 people from 2014 to 2019, respectively [12]. Similarly, there was a higher incidence rate of dengue fever in Southeast Asia; around 297.31 cases per 100,000 people in 2019, including in Cambodia, Laos, Malaysia, Philippines, Singapore, Thailand, and Vietnam [13]. Chikungunya fever is another MBD with a large number of outbreaks and cases in Southeast Asia. From 2005 to 2008, chikungunya fever was reported to have more than 1.3 million cases in almost every Indian state [14]. It was estimated that India had the largest estimated burden of JE in 2019, with 22,219 JE cases [15]. As for mosquito species in China, there were 302 mosquito species, 17 genera according to Fauna Sinica, Insecta Vol. 8, Diptera: Culicidae Ⅰ and Fauna Sinica, Insecta Vol. 9, Diptera: Culicidae Ⅱ, published in 1997 [16,17]. Up to now, the amount of mosquitoes in China was updated to 377 species and 18 genera according to China Species Library, the first authoritative database of all Chinese biological species [18]. In addition, urbanization, climate change, bird migration, and the cross-continent movement of humans and animals inevitably affected the distribution of mosquitoes and arboviruses [19,20]. Therefore, in-depth research into mosquitoes and arboviruses is demanded with their ongoing geographic expansion. This will help to update the spatial distributions of both mosquitoes and mosquito-borne arboviruses, as well as their underlying risk determinants, in a timely fashion.

In other countries, the distribution of mosquito species was studied and determined based on an enormous field investigation. According to the book Distribution and Habitats of Japanese Mosquitoes (Diptera: Culicidae) 2nd ed., there are around 112 mosquito species in Japan, divided into two groups; 40 species of northern mosquitoes and 72 species of southern mosquitoes [21]. In some countries with limited areas, national mosquito species surveys are easier to conduct, such as Seychelles with 22 mosquito species [22] and Mali with 106 species [23]. However, in China, the complex ecological and topographic environments such as diverse host animals and land cover, together with the changing behavior of a huge population such as accelerated urbanization and increasing human movement, create both opportunities and challenges for studying the distribution and ecology of mosquitoes and mosquito-borne arboviruses [24]. A recent study compiled the distribution data of mosquito-associated viruses and their mosquito vectors (2428 records) from 283 literatures reported during 1957 and 2019 [25]. Another study reviewed the geographic distribution and genomic sequencing of recognized mosquito-borne viruses in the mainland of China from 1910 to 2010 [26]. Zhang et al. assembled the geographical distribution of *Anopheles*, the sole vector for malaria, around China [27]. However, none of these studies mapped the distributions of all predominant mosquito species and arboviruses that have been found in China, nor did they systematically investigate ecological niches of either major mosquito species or arboviruses associated with high disease burdens in China at a fine geographic resolution.

Here, we conduct an up-to-date review on the spatial distributions of predominant mosquito species and arboviruses detected from mosquitoes, animals, and MBD patients in the mainland of China. Based on the assembled data, we build predictive models to assess the contributions of relevant socioenvironmental factors to the ecological suitability of 26 selected mosquitoes and two mosquito-borne arboviruses with the highest disease burden in China, and we subsequently map model-projected risks to inform future surveillance and control efforts.

## 2. Materials and Methods

### 2.1. Data Sources

Three major data sources were used to extract known records of mosquito species and mosquito-borne arboviruses that had been reported in the mainland of China between January 1954 and March 2021: (1) the reported incidences of notifiable MBDs from the Chinese Scientific Data Center for Public Health, (2) literature review (Appendix A), and (3) the location data of arboviruses from GenBank that is the National Institutes of Health (NIH) genetic sequence database, an annotated collection of all publicly available DNA sequences [28].

Clinically diagnosed or laboratory-confirmed dengue fever and JE cases at clinics and hospitals are reported, as mandated by the Ministry of Health, to the Chinese Information System for Diseases Control and Prevention. The county-level data of human cases of identified origin for dengue fever and JE during 2014–2018 were collected from the Chinese Scientific Data Center for Public Health and were used in the models for arboviruses. As for the literature review, five main electronic databases (PubMed, ISI Web of Science, China WanFang database, China National Knowledge Infrastructure, and Chinese Scientific Journal Database) were searched for studies published between January 1954 and March 2021, using the following keywords: “mosquito” or “mosquitoes”, “mosquito-borne virus” or “mosquito-borne disease”, and “China” in English and Chinese (Appendix A). We also checked the references in retrieved articles to reach more relevant articles. Each article was carefully reviewed by two team members independently to collect the following information using a standard form: time and location of investigation, type of samples, spatial resolution, mosquito or arbovirus species identified, laboratory test methods, number of samples to test, and test results for arboviruses. Any disagreement between the two staff members was resolved by discussion and consensus among the reviewers and other co-authors. Only studies with clearly identifiable results, i.e., presence or absence, time and location of mosquito species or arboviruses, were included in our database. For articles containing ambiguous data, the original authors were contacted for clarification; if the ambiguity was not clarified, the data in question were excluded from our database. We excluded records of mosquito species that were detected only in prefectures and provinces. As a limitation to the literature access, our study focused on the geographical scope of the mainland of China; some mosquitoes exclusively found in Taiwan, Hongkong and Macao were not included in our database.

Only viruses capable of transmitting to humans were included in this study, while insect-specific viruses (e.g., Yichang virus) and viruses that transmit only to animals yet not to humans (e.g., Akabane virus) were excluded. In addition, newly identified viruses in the past 20 years that have not yet been detected in humans, were also included as possible disease-causing viruses, such as Menghai rhabdovirus (MRV). The arboviruses included in our database were detected from mosquitoes, host animals, and humans by any of the following laboratory tests: reverse transcription polymerase chain reaction (RT-PCR), PCR, isolation, or culture. For humans and host animals, cross-sectional studies that specific antibodies detected in the blood sera collected were also included in our database, only for displaying the probable distribution of them. The positive detection of specific IgM antibodies was defined as a probable case of arboviruses. In addition, the database of GenBank contains information on the collection time and location of strains of arboviruses. Thus, we also extracted the following information from GenBank: accession, time and location of collection, and serotypes or genotypes of arboviruses. In total, 1995 records were extracted from GenBank for the current study. The data from the literature and GenBank were integrated to form one database down to the county level for final analyses.

For the purpose of ecological modeling, in total, 53 factors (18 environmental, 19 ecoclimatic, 7 social and 9 biological factors) that are potentially associated with the ecology of mosquito species and MBDs were collected at the county level (Appendix A) [24,29,30]. The choice of variables is mainly based on empirical ecological evidence in the literature and their spatial variability. In addition, we focus on ecological variables that are potentially shared by multiple species so that the results can be compared across species. Thirty-eight years’ (1981 to 2018) worth of climatic data were collected from 2006 weather surveillance stations in the mainland of China, covering 74.0% of 1228 surveyed counties. The climatic data include average monthly meteorological variables such as temperature, maximum temperature, minimum temperature, and rainfall over 38 years. For the 877 counties (319 with mosquito presence) without weather stations, the mean values of the nearest five surveillance stations were used as proxies for their meteorological variables. From these longitudinal meteorological variables, 19 cross-sectional ecoclimatic variables (BIO01−19, also called bioclimatic variables recommended by the U.S. Geological Survey) were calculated, and their yearly averages were used as predictors in our machine learning models [31]. These ecoclimatic variables better capture the seasonal trends of different species related to their physiological constraints than traditional meteorological variables, and have been widely used in ecological studies [31]. Some of these ecoclimatic predictors are highly correlated. We performed a clustering analysis on these predictors based on their pairwise correlations using the R package “NbClust”. Specifically, a binary distance matrix was formed, with the distance between any pair of ecoclimatic variables being 0 if the absolute value of the correlation coefficient is bigger than 0.8, and 1 otherwise. The best number of clusters was chosen by the Krzanowski and Lai index [32]. This clustering analysis found eight clusters of the ecoclimatic variables (Appendix A). A continuous distance matrix where the distance is one minus the absolute value of the correlation coefficient also identified the same clusters. Only one predictor from each group was used for model-fitting.

China has updated its land cover data every 5–10 years since 1980. Raster-type land cover data of China in the years 1980–2015 were obtained from the Resource and Environment Science and Data Center. We used the average value of land cover data in 1980–2015 for the models of mosquito species and arboviruses. Livestock density data in 2010 were obtained from the Food and Agriculture Organization of the United Nations. The mammalian richness in 2013 was collected from the Socioeconomic Data and Applications Center. The population data at the county level were derived from the Sixth National Census of China in 2010. Social data, including the numbers of hospitals, clinics, and emergency rooms, percentage of female population, and percentage of residents ≥60 years old were provided by the National Bureau of Statistics. These predictors at the county level were extracted and calculated from these data using the ArcGIS Desktop 10.7.0.10450 software (ESRI Inc., Redlands, CA, USA) (Appendix A). Data cleaning and reorganization with regard to these variables were performed in the R v4.0.3 (R Core Team, 2020).

### 2.2. Spatial Mapping

Occurrences of mosquitoes, arboviruses, and human cases were geo-referenced at the county level whenever feasible, or at the prefecture or province level otherwise. All the records were determined by the location of investigation at the county level and mosquito or arbovirus species identified, without considering the sampling time. In total, 80.2%, 60.4%, and 53.3% of the occurrence records of mosquitoes, arboviruses, and MBDs were geo-referenced at the county level. All maps were produced using the package “sf” of the R 4.0.3 software, and the digital map of China’s administrative divisions was downloaded from the Resource and Environment Science and Data Center, Chinese Academy of Sciences.

### 2.3. Ecological Modeling

Twenty-five predominant mosquito species with recorded occurrence in ≥80 counties plus *Ae. Aegypti*, which is the competent vector for DENV and other arboviruses, were included in the modeling. In the current study, a case control study design and the boosted regression tree (BRT) models were used to establish the classification of the presence or absence of each of the 26 major mosquito species at the county level [33,34]. We divided all counties where mosquitoes were surveyed into “case” and “control” by whether they had the occurrence records or not for specific mosquito species, and estimate the effect of different predictors and the presence probability of mosquitoes by the BRT models. At the first step, based on whether all referred mosquito species were identified at the collection location, we classified each record into two types of field investigation (complete investigation and incomplete investigation). For each given mosquito species, counties with at least one record of occurrence were considered as “cases”, and those conducting complete investigations yet lacking any evidence of occurrence were considered as “controls”. The numbers of “cases” and “controls” for each mosquito species were listed in Appendix A. The remaining counties where mosquito surveys have not been conducted or have not yielded conclusive findings were excluded from model building, but were included for risk mapping. For example, among a total of 1228 counties where any mosquito species were surveyed, 555 counties recorded the occurrence of *Ae. albopictus*, and were thus considered as “cases”, 386 counties with complete investigations, yet no existence evidence of *Ae. albopictus* considered as “control” sites, as well as the remaining 1917 counties (287 counties with incomplete investigation yet no existence evidence of *Ae. albopictus*), were defined as missing and excluded for modeling the distribution of *Ae. albopictus*.

A BRT model at the county level was fitted to the training set to assess the contributions of ecoclimatic, environmental, social and biological predictors to the geographic distribution of the given mosquito species. The BRT model is a popular approach to ecological studies and has been widely used for the risk mapping of infectious diseases such as avian influenza, rabies, and helminth [35,36,37,38]. The BRT model couples the advantages of two algorithms, regression trees and machine learning techniques, and allows nonlinear relationships between outcomes and covariates, and multicollinearity among covariates [39]. For each BRT model, 35 variables, including 18 environmental, 8 ecoclimatic, and 9 biological factors were used as potential predictors (Appendix A). The fitted model was used to project risk levels in counties without mosquito surveys [40,41]. To counterbalance the potential sampling bias of survey counties, we built a logistic regression model for the selection of mosquito survey counties, with all ecoclimatic and socio-environmental variables as predictors (Appendix A). The response of this model was one for all mosquito-surveyed counties and zero for unsurveyed counties. The predictors were chosen using a backward procedure at the significance level of 0.05. The reciprocals of predicted sampling probabilities of all surveyed counties were first rescaled to have a mean of one, and then used as weights in the BRT models for the 26 major mosquito species [42,43,44].

A tree complexity of five, a learning rate of 0.005, and a bagging fraction of 75% were used for the primary analysis, based on their satisfactory performance in our previous research [45,46,47]. Bagging is a procedure that resamples data points to fit sequential trees, so as to improve predictive performance. A 10-fold cross-validation was used to identify the optimal number of trees using the gbm.step function in the R package “dismo”. The output of a BRT model consists of both predicted probabilities of occurrence and relative contributions (or influences) of predictors. The relative contribution (RC) is calculated based on how many times a predictor is chosen for splitting, and how much each split improves the objective function, averaging over all trees. These RCs of all predictors are standardized so that they add up to one [39]. A two-stage bootstrapping procedure was employed to provide a more robust and parsimonious estimation of model parameters. In each stage, the following split-and-fit step was repeated a certain number of times. A training set with 75% data points was randomly selected by bootstrapping without replacement, and the remaining 25% served as a test set. A BRT model was built using the training set, and then applied to the test set for validation if needed. In the first stage, the split-and-fitting step was repeated ten times to screen important predictors. The validation of the trained model using the test set was not performed at this stage. Predictors that had a RC <2% for all bootstrap training sets were excluded from the next stage. In the second stage, the split-and-fitting step was repeated 100 times using the remaining predictors. As no variable selection was made in this stage, all 100 models had the same predictors but yielded different contribution estimates. The RCs of the predictors were averaged over the 100 BRT models to represent their final RCs. The receiver operating characteristic (ROC) curves and areas under the curve (AUC) based on the test sets were also averaged to represent the final predictive performance. The standard deviations and 95% percentiles of the RCs and AUCs across the 100 models were used to quantify the uncertainty in the estimation. Considering that there could be false negative and false positive counties in the observed data, we also calculated partial area AUC with a tolerance level of 0.2 for omission error [48]. For partial area AUC, the horizontal axis is the total rate of positives rather than false positives. We presented the ratio of the partial AUC to the area under the random selection line (diagonal line), as suggested by Peterson et al. [48]. Finally, the predicted probabilities were averaged over the 100 models to represent the final estimates of the county-specific probabilities of presence, which were mapped for the 26 major mosquito species [35,37,40,41,47]. BRT modeling was conducted using the R packages “dismo” and “gbm”, and predictive performance was assessed using the R packages “ROCR” and “pROC” in the R v4.0.3 (R Core Team, 2020).

We also performed a sensitivity analysis using a learning rate of 0.01 for selected mosquito species, but found no substantial difference in the contribution estimates. Due to both the data size (35 predictors) and the number of model runs ([26 mosquitoes] × 100), we cannot afford a full cross-validation optimization for all model configuration parameters. To determine model-predicted high-risk counties for each mosquito species, we chose a cut-off value that maximizes sensitivity + specificity along the ROC curve for each final BRT model [49,50]. Counties with predicted probabilities above the cut-off value for a given model were considered as having a high risk of harboring the corresponding mosquito species. For each mosquito species, the number, area, and population size of model-predicted high-risk counties were compared to the quantities of counties with observed occurrences (Table 1).

### 2.4. Clustering Mosquitoes with Similar Ecological Niches and Their Spatial Distribution

To explore similarity in ecological niches among the 26 predominant mosquito species, a hierarchical cluster analysis based on the weighted-average linkage method was performed [51]. Features used for clustering were formed as the following. We first excluded predictors that are not influential (excluded from final models) for all 26 mosquitoes. For each mosquito species, three quantities associated with each remaining ecological predictor were calculated as features for clustering. One quantity is the average RC of this predictor in the final 100 BRT models. If the predictor was not included in the final models for this mosquito species, its RC was set to zero. The second quantity is a measure for the difference in this predictor between case counties (positive for the given mosquito species) and all counties. We first calculated the median value of this predictor among all case counties and quartile intervals of the predictor among all counties in the nation. We then assigned one of the numbers 1–4 according to which quartile interval the median lies in, e.g., assign 1 (4) if the median lies in the lowest (highest) quartile. The third quantity is the linear correlation between the predictor and model-predicted presence probabilities of the given mosquito species among all counties (averaged over the 100 models). These three quantities of all ecological predictors jointly serve as features for clustering. A dendrogram was created to demonstrate the clustering pattern of these 26 mosquito species, together with a thematic matrix illustrating the features (Figure 1). This matrix has mosquito species as rows and predictors as columns. The color of each cell in the matrix shows the average RC and the number shows the quartile (1–4 for 1st–4th quartiles) location of the median of cases. To map geographic distributions of the identified clusters of mosquito species at the county level, we define the presence of each cluster as the presence of any mosquito species in that cluster.

### 2.5. Population at Risk for Main MBDs with High Disease Burden

Two-stage generalized boosted regression tree (GBRT) models were built at the county level to explore potential drivers for the presence and incidence of DENV and Japanese encephalitis virus (JEV) infection, two of the most commonly reported mosquito-borne arboviruses in China. GBRT is the generalized version of BRT to handle outcomes with special distributions, such as binomial, Poisson, Gamma, etc., in which the loss function is related to the likelihood, and we used the version implemented in the R package “xgboost” [52]. Due to disability to represent the risk of disease, the imported cases were excluded, and the logarithmic transformed number of imported dengue cases (index of case importation) was included in the model of DENV as a potential predictor. For the presence and absence of the disease, all counties where the viruses were detected in mosquitoes and host animals according to literature or human cases of the associated MBDs reported by surveillance were regarded as presence. For each arbovirus, in addition to the same 35 potential predictors, additional predictors, including 8 social factors and the present possibility of 6 vector mosquitoes, were included in GBRT models (Appendix A). At the first stage, a logistic GBRT model was used to fit the presence/absence of each virus by county. Any record of positive detection from human cases, host animals, or mosquitoes was regarded as the presence of the virus at the county level, otherwise absence was assigned to the county. This stage accounts for the excessive amount of zero case numbers in the majority of the nation. At this stage, similar to the BRT model for mosquitoes, the cut-off values were calculated to estimate the presence of viruses. At the second stage, a GBRT model was fitted to counties with non-zero average annual incidences of reported human cases from 2014 to 2018, assuming that the incidence rates follow a gamma distribution. The gamma distribution was chosen because it best fits the observed non-zero average annual incidences. At both stages, the models were run with a learning rate of 0.05, a max tree depth of 8, and a bag fraction of 0.75. The best number of trees (nrounds in Xgboost) was chosen by 5-fold cross validation. Similar to the ecological models of mosquitoes, we first included all of the respective corresponding predictors shown in Appendix A for the two arboviruses, and modeling was repeated ten times, and then excluded predictors with RCs <1.5% in the final model. The final model fitting process was repeated 100 times, and the final results (estimated RCs and response curves) are averaged over the 100 models. The model-fitted incidence was calculated as the following: if the stage-1 logistic-model-predicted probability of presence was less than the cut-off value, then the predicted incidence was set to 0; otherwise, the predicted incidence was set to the probability of presence times the mean incidence predicted by the stage-2 gamma model. The GBRT models were fitted using the package “xgboost” in the statistical platform R v4.0.3 (R Core Team, 2020).

## 3. Results

Through the literature review, we found a total of 36,731 references: 5750 in English, and 30,981 in Chinese, which met our search criteria. With a consensus of two independent reviewers, 2064 publications met our study inclusion criteria, from which 1142 reported detections of arboviruses were extracted. After pooling the data from all sources, we obtained an integrated database of 13,782 mosquito records. The 26 predominant mosquitoes belong to 4 genera. We assembled 5132, 3762, 3349, and 800 occurrence records of *Culex*, *Anopheles*, *Aedes*, and *Armigeres* respectively, the four most prevalent mosquito genera.

### 3.1. Distribution of Mosquito Species in the mainland of China

We compiled a database on the geographic locations of 339 known mosquito species from 18 mosquito genera, together with 35 mosquito-borne arboviruses detected in mosquitoes, animals, or humans, across 1228 counties (43% of all counties) in the mainland of China (Appendix A). The most widely distributed mosquito genus is *Anopheles* (in 898 counties), followed by *Culex* (830), *Aedes* (799), and *Armigeres* (383) (Appendix A). At the species level, *Ae. albopictus*, *Anopheles* (*An.*) *sinensis*, and *Culex* (*Cx.*) *tritaeniorhynchus* were each identified in >500 counties (Appendix A). The diversity of mosquito species differed substantially across the seven biogeographic zones, which are defined by climatic and ecological characteristics (Appendix A). Mosquito species are most diverse in the South China district, Central China district, and Southwest district, hosting 229, 188, and 148 mosquito species, respectively (Appendix A). Thirteen prefectures reported ≥60 mosquito species, including five in Southwest China, four in Central China, and four in South China (Appendix A), most below a latitude of 30^°^ north. In contrast, the Qinghai-Tibet district has the lowest mosquito diversity (Appendix A).

### 3.2. Ecological Modeling and Clustering of Predominant Mosquito Species in China

Ecological modeling was performed for 26 mosquito species, including 25 predominant mosquito species recorded in ≥80 counties, plus *Ae. aegypti* which is the competent vector for DENV and other arboviruses. All 26 mosquito species had yielded highly predictive models, with the average testing AUC ranging from 0.68 to 0.95 (Table 1) and the testing partial AUC ratio ranging from 1.13 to 1.94 (Appendix A). The model-predicted high-risk areas for all of the 26 mosquito species are much more extensive than what has actually been observed, 89–651% of the number of affected counties, 60–664% of the geographic area coverage, and 70–489% of the population size with residence (Table 1). Based on the predictive model, *An. sinensis* and *Cx. tritaeniorhynchus* remained with the widest geographic distributions, as had been observed, potentially affecting 978 million and 1043 million people with residences within 1930 and 1924 counties, respectively (Table 1). *Cx. pipiens pallens*, *Ar. subalbatus*, and *Ae. albopictus* might affect more than 700 million people. At the scale of 3.3–5.8 million km^2^, *An. sinensis*, *Cx. vagans*, *Cx. modestus*, *Ae. vexans* and *Ae. dorsalis* are the top five mosquito species affecting the largest areas (Table 1). Based on these models, temperature seasonality (defined as a standard deviation of monthly temperature) and annual cumulative precipitation are the two most important drivers, with an average RCs ≥5% in the ensembled models for 23 and 17 mosquito species, respectively. Mammalian richness had demonstrated an average RCs ≥5% for 15 mosquito species (Appendix A). These predictors, however, have exerted differential impacts among the studied mosquito species, even for those in the same genus (Appendix A). For example, a high level of temperature seasonality was associated with a high probability of presence for *Ae. vexans* and *Ae. Dorsalis*, but with a low probability for *Ae. albopictus* and *Ae. aegypti* (Appendix A).

Based on the determined ecological predictors, the 26 mosquito species are grouped into seven clusters with similar predictors, which also showed a clear pattern of spatial aggregation (Figure 1). Clusters I–II are mainly located in northern China and meteorologically featured by low temperature, high degree of temperature seasonality, and low annual cumulative precipitation (Appendix A). By contrast, Clusters III–VII are mostly distributed in zones V–VII in southern China and featured by high temperature, low degree of temperature seasonality, and high annual cumulative precipitation, with an exceptionally high effect observed for Cluster III. Similar landcover features were also presented within the clusters. For example, Clusters IV–VI were featured by high coverage of paddy fields and fewer rainfed fields, and Clusters III–IV are characterized by more forests but fewer lakes and reservoirs. Three mosquito species, *An. pattoni*, *Cx. pipiens pallens*, and *Ae. aegypti*, have their own unique ecological niches, and are thus not clustered with others. In terms of geographic distribution, however, *An. pattoni*, *Cx. pipiens pallens*, and *Ae. aegypti* are close to Clusters I, II, and III, respectively.

### 3.3. Mosquito-Borne Arboviruses Known to Infect Humans

Excluding viruses without any evidence for human infection, we collected data on 35 mosquito-borne arboviruses, among which 31 are known to infect humans and the remaining four emerged in the past 20 years and are considered likely to infect humans (Figure 2 and Appendix A). *Cluex* is the genus harboring the highest number of arboviruses (22 viruses), followed by *Anopheles* (13), *Aedes* (10), and *Armigeres* (7). At the species level, *Cx. tritaeniorhynchus* harbors the highest variety of arboviruses that included 19 known viral species. Other mosquito species that carry more than five viruses are *An. sinensis* (11 viruses), *Cx. pipiens quinquefasciatus* (9), *Ae. albopictus* (7), and *Ae. aegypti* (6).

*Flavivirus* was the genus that parasitized the highest variety of mosquitoes (28 mosquitoes), followed by *Alphavirus* (15). Within the *Flavivirus* genus, JEV and DENV were the most frequently recorded species that were carried by 27 and 7 of the studied mosquitoes. JEV is detected primarily in *Cx. tritaeniorhynchus*, while DENV is detected primarily in *Ae. albopictus* and *Ae. aegypti* (Figure 2). Other viruses that parasitize more than five mosquito species are Getah virus (GETV) in the *Alphavirus* genus (12) and Banna virus (BAV) in the *Seadonavirus* genus (8).

In addition, IgM responses to Barmah Forest virus (BFV) and Ross River virus (RRV) have been identified in patients, but the arboviruses have not been detected in mosquitos yet. In contrast, 12 arboviruses were found in mosquitoes, yet no human cases have been reported, including Murray Valley encephalitis virus (MVEV), Sagiyama virus (SAGV), Mayaro virus (MAYV), Western equine encephalitis virus (WEEV), Batai virus (BATV), Oya virus (OYAV), Tibet orbivirus (TIBOV), Mangshi virus, Kadipiro virus (KDV), Longchuan virus, MRV, and Nam Dinh virus (NDV).

#### 3.3.1. The Locations of Arboviruses from Human Cases and Mosquitoes

In the *flavivirus* genus, DENV and JEV were two of the most widely distributed mosquito-borne arboviruses in the mainland of China, identified in provinces. We mapped the distribution of genotypes of JEV and serotypes of DENV detected from human cases, animals, and mosquitoes in China (Appendix A). The genotypes of JEV isolated in China are mainly GI and GIII genotypes. A total of 49 strains of JEV were isolated from JE cases, including 17 strains of GI and 32 strains of GIII. In total, 78 strains of JEV isolated from host animals were identified by nucleic acid sequencing, including 26 strains of GI and 52 strains of GIII. In addition, genotypes of JEV from 908 pigs were identified based on OD450 value, including 444 viruses of GI and 297 of GIII. A total of 785 strains of GI and 130 strains of GIII were isolated from mosquitoes, of which GI was mainly found in Zhejiang (229 strains), Yunnan (174), and Shandong (137) provinces, while GIII was mainly found in Yunnan Province (53). In addition, one strain of JEV of GV was isolated from *Cx. tritaeniorhynchus* in the Tibet Autonomous Region.

DENV comprises four serotypes, namely DENV-1, 2, 3, and 4. A total of 1238 imported dengue cases were identified for serotypes, mainly in Guangdong (534 cases) and Yunnan (320) (Appendix A). Among them, DENV-1 accounted for 51.3% (635 cases), followed by DENV-2 (27.0%, 334 cases), DENV-3 (13.4%, 166 cases), and DENV-4 (8.3%, 103 cases). The composition of local cases (totally 5749 cases identified) and cases of unknown origin (totally 20,083 cases, mostly local cases) was similar to that of imported cases, with DENV-1 predominating (68.2%, 17,616 cases), followed by DENV-2 (23.2%, 5997 cases), DENV-3 (7.1%, 1837 cases), and DENV-4 (1.5%, 382 cases). Five provinces reported more than 1000 local or unknown origin cases, including Guangdong (18,097 cases), Yunnan (2590), Hainan (1079), Guangxi (1575), and Zhejiang (1499) provinces, with different dominant serotypes. DENV-1 was the dominant type in Guangdong, Yunnan, and Guangxi provinces, while DENV-2 was predominant in Zhejiang and Hainan provinces. Overall, 115 strains of DENV in mosquitoes were isolated, among which DENV-2 dominated (51 strains), with DENV-1 mainly isolated in Guangdong Province, DENV-2 and 3 mainly found in Hainan Province, and DENV-4 mainly found in Yunnan Province.

West Nile virus (WNV) is the third most widely distributed virus, which was detected from mosquitoes in Xinjiang and Yunnan provinces, followed by Tembusu virus (TMUV) detected from mosquitoes in Shandong, Yunnan, and Guangxi provinces, by nucleic acid testing (Figure 3A). Yellow fever virus (YFV), Zika virus (ZIKV), and MVEV were detected from mosquitoes in Yunnan Province by nucleic acid testing or isolation. Probable cases with positive anti-WNV IgM were reported in eight provinces by enzyme-linked immunosorbent assay (ELISA) or indirect immunofluorescence assay (IFA), with more cases reported in the Xinjiang Autonomous Region [53,54]. In total, 63 cases infected with TMUV were reported by RT-PCR in Shandong Province. A total of 32 imported cases with ZIKV infection were reported in 2016–2019, mainly in Guangdong Province, among international travelers. Notably, five probable local ZIKV cases were reported by ELISA in Guangxi Autonomous Region. In 2016, a total of 11 cases infected with the YFV were imported from Angola.

CHIKV, as the most commonly reported alphavirus, was mainly determined in coastal and border provinces in the south and southwest regions, such as Guangdong and Yunnan provinces (Figure 3B). From 2010 to 2019, by nucleic acid testing, there had been four local outbreaks of CHIKV in China, two in Guangdong Province during 2010 (129 and 7 cases) [55,56], one in Zhejiang Province during 2017 (3 cases) [57], and one in Yunnan Province during 2019 (23 cases) [58]. In addition, a large number of imported cases and detections of CHIKV from mosquitoes were reported in the same regions by nucleic acid testing and isolation (Figure 3B). Sporadic cases infected with Sindbis virus (SINV) were reported in Yunnan, Zhejiang, and Xinjiang provinces, and SINV was detected in mosquitoes in Yunnan Province by nucleic acid testing and isolation. GETV had the most widespread geographic scope in the mosquitoes among alphaviruses, but probable cases were only reported by ELISA in Henan province.

Human probable cases with Ťahyňa virus (TAHV) infection, a member in the genus of *Orthobunyavirus*, were found by IgM tests in Xinjiang and Qinghai provinces (Figure 3C). Ebinur Lake virus (EBIV), a novel orthobunyavirus, was first detected in *Cx. modestus* in the mainland of China in 2012, and 17 probable cases were identified via IgM tests in the Xinjiang Autonomous Region in 2014 [59]. In the genus of *Orbivirus*, Yunnan orbivirus (YUOV) and novel orbivirus (NOBV) were found in both mosquitoes by nucleic acid testing and isolation and probable cases by ELISA in Yunnan Province, while TIBOV was detected only in mosquitoes by nucleic acid testing and isolation in several provinces (Figure 3D). BAV, a seadonavirus, was detected from probable human cases by ELISA in Yunnan Province, although its presence in mosquitoes has shown a much wider geographic scope (Figure 3E). Two other seadonaviruses, Mangshi virus and KDV, were only found in mosquitoes in Yunnan Province by nucleic acid testing and isolation.

One confirmed case of Rift Valley fever virus (RVFV) by PCR, a phlebovirus mainly found in Africa and the Arabian Peninsula, was imported to Beijing from Angola in 2016. Longchuan virus, a novel quaranjavirus, was found in *Cx. pipiens quinquefasciatus* by sequencing in 2015 in Longchuan County, Guangdong Province. MRV, a member of rhabdoviridae, has been reported in mosquitoes in Yunnan and Gansu provinces. Meanwhile, NDV, a Mesoniviridae first found from mosquitoes in Vietnam [60], was detected in mosquitoes by nucleic acid testing and isolation in Yunnan and Guangdong provinces (Figure 3F).

#### 3.3.2. The Locations of Arboviruses from Serological Investigation of People

There were abundant detections of specific antibodies for mosquito-borne arboviruses from people by ELISA, IFA, and other serological methods in China (Appendix A). Except for the notifiable diseases, JE and dengue fever, a total of six *flavivirus* species were detected for their antibodies in China. Anti-WNV antibodies was detected in local residents in Yunnan and Xinjiang provinces with detection rates of 2.08–50.00%, and in international travelers in Guangdong Province with 31.82% and 36.52%. Among 273 residents in Guangxi Province, 21 was reported with anti-ZIKV IgG antibodies. Among 132 people who worked on duck farms in Shandong Province, 95 were detected to have anti-TMUV antibodies. The other three flaviviruses, KUNV, MVEV and SLEV, were all reported in Yunnan and Guizhou provinces, with detection rates of 5.33–35.42%, 7.47–54.22%, and 1.19–2.13%, respectively.

As for *Alphavirus*, there were 11 species detected with specific antibodies in China. Anti-SINV antibodies were detected in the serum of local people in most of China, with positive rates of 0.36–59.15%. In particular, in Yunnan Province, 36 serosurveys were conducted with positive rates of 0.89–59.15% of anti-SINV antibodies. Anti-CHIKV antibodies were detected not only in local residents, but also in international travelers. In Yunnan and Guangdong provinces, international travelers were detected with positive rates of 21.43% and 4.44–5.19% for anti-CHIKV antibodies, respectively. Similarly, local residents with anti-CHIKV antibodies were reported in southwestern and southern China, mainly in Yunnan Province, with 35 serosurveys and positive rates of 0.31–43.78%. Local people were reported with positive rates of 1.85–26.37% for anti-GETV IgG, mainly in southwestern and southern China. Anti-RRV antibodies were found in Hainan and Guizhou provinces, with positive rates of 1.02–8.70%. In a serosurvey of BFV in Guizhou Province, people were detected with positive rates of 1.49%. The serosurveys of other alphaviruses were conducted in most of China, especially in Yunnan and Hainan provinces.

Three orthobunyaviruses, TAHV, BATV, and SSHV, were found with the detection of antibodies in local residents. Similar to the location of TAHV cases, anti-TAHV antibodies were detected in Xinjiang and Qinghai provinces, with positive rates of 0.28–18.27%. Anti-BATV antibodies were found in Yunnan Province with positive rates of 4.17–5.00%. In Yunnan, Hainan, and Guangdong provinces, SSHV was detected in local residents with positive rates of 2.13–26.47%. In addition, in Yunnan Province, antibodies against four arboviruses of other genera were detected in local residents; in detail, BAV with positive rates of 1.72–9.68%, NDV with 2.15–15.05%, NOBV with 2.75–3.70%, and YUOV with 5.93–6.59%.

#### 3.3.3. The Locations of Arboviruses from Animals

We also georeferenced and mapped locations of the positive detections of 9 arboviruses from host animals by nucleic acid testing and isolation in China, including 4 *Flavivirus* species, 3 *Alphavirus*, 1 *Orthobunyavirus*, and 1 *Orbivirus* (Appendix A). Four flaviviruses, JEV, DENV, WNV, and TMUV, were detected from multiple animals. JEV was the most widely distributed among animals, primarily pigs, in most of China except the northwestern region. The second widely distributed flavivirus is TMUV, which was found in three kinds of poultry, ducks, chickens, and geese, as well as two wild birds, sparrows, and pigeons, in eastern and southeastern coastal areas of China where duck farms are widely distributed. DENV was detected by isolation and RT-PCR from bats in Hunan Province. WNV was detected by RT-PCR from two wild birds in Beijing City.

Altogether, 3 species in the genus of *Alphavirus*, GETV, CHIKV, and RRV were detected in multiple animals. GETV was widely distributed in animals in northern, central, and southeastern China, especially in pigs. In Hainan Province, CHIKV and RRV were found from bats by isolation. BATV, an orthobunyavirus, was detected from ducks in Zhejiang Province by RT-PCR, and cows in the Inner Mongolia Autonomous Region by isolation and PCR. TIBOV was found in cows in Yunnan Province by RT-PCR. In addition, abundant detections of antibodies against 15 arboviruses showed a wider distribution in geography.

### 3.4. Modelling for Geographic Distribution of Human Cases with DENV and JEV

We made further efforts to predict the geographic distribution of DENV and JEV, two of the most widely distributed arboviruses in the mainland of China. According to the national reportable data in 2014–2018, the majority of human dengue cases were reported in seven counties in Southern China, with an annual incidence of 0.062–0.101 per 100,000 persons (Figure 4A). The model-predicted high-risk areas are more spatially aggregated than the observed ones, but the overall distributions are largely similar, covering 128 thousand square kilometers (Figure 4B). Approximately 74.2 million people reside in high-risk areas, and the model-predicted incidence rates of dengue exceed 0.250 per 100,000, among which 18 counties had significantly higher model-predicted incidence than reported incidence (<0.100 per 100,000). The mean temperature of the warmest quarter was the leading contributing factor to the presence of DENV (RC ≥ 34%). Other drivers with RC ≥5% include annual mean temperature, annual precipitation, model-predicted presence probability of *Ae. albopictus*, the number of general hospitals, and population density (Table 2). Among the counties with dengue cases reported, annual temperature, mean temperature of the wettest quarter, annual precipitation, index of case importation, and the model-predicted presence probability of *Ae. albopictus* were important predictors (RC ≥5%) for the average incidence rate of human dengue cases (Table 2 and Appendix A).

Human cases of JEV were primarily clustered in southwestern, central, and northern China, coinciding with model-predicted high-risk areas (Figure 4C,D). JEV was detected from mosquitoes (mainly *Cx. tritaeniorhynchus*) and animals (mainly pigs) in a broader area that extended to mountainous northeastern provinces and the coastal provinces in the east, in comparison with regions that report human cases (Figure 4C). Approximately 59.6 million people reside in high-risk areas covering 465 thousand square kilometers, where the model-predicted incidence rates of JE exceed 0.221 per 100,000, among which 12 counties had significantly higher model-predicted incidence than the reported incidence (<0.150 per 100,000). Annual temperature, temperature seasonality, precipitation of the driest quarter, mammalian richness, and the presence possibility of *Cx. tritaeniorhynchus* were the important factors for the presence of JEV with all the RC ≥ 5%. Among counties with reported JE cases, the incidence of JE was affected by the coverage of grasslands with the greatest contribution (RC = 24.0%), followed by a smaller effect on the density of chicken (RC = 16.3%). Other important predictors with RC ≥ 5% included the mean temperature of the warmest quarter, precipitation of the driest quarter, and densities of pig and duck (Table 2; and Appendix A).

## 4. Discussion

We have endeavored to assemble a comprehensive set of occurrence records of mosquito species and mosquito-borne arboviruses that can potentially infect humans in China covering a time span of 70 years. We reported the locations of mosquito species and arboviruses at the county level (Appendix A). Our refreshed data on the mosquito species and related arboviruses provided a more complete list than previous studies, e.g., we identified 27 mosquito species harboring JEV and 19 arboviruses detected in *Cx. tritaeniorhynchus*, compared to the previously reported numbers of 23 and 13, respectively [25]. This finding, on the other hand, stressed the necessity of persistent surveillance of the arboviruses. Using ecological models based on a machine learning algorithm, we found that the geographic scopes of predominant mosquito species (particularly *Cx. vagans* and *Cx. mimulus*) could be up to 7-fold as large as what has been observed, likely due to limited field investigations or incomplete sampling (Table 1). However, it is also possible that our models were underspecified and thus overestimated the scopes.

Similarities in meteorological features and land area were both correlated with similarity in ecological niche, and therefore the mosquito and mosquito-borne arboviruses assemblages overall. Here, we grouped mosquito species by their favorable ecological characteristics other than from their genera. We found seven clusters of mosquito species that share similar ecological niches and geographic distributions, and the key ecological predictors may differ, even within the same genus. Therefore, the clustering patterns provided assemblage-level ecological factors, which might help facilitate risk assessment and field investigation in a more efficient way than their separate evaluation.

The most powerful ecoclimatic factors that favored enhanced mosquito activity are a lower degree of temperature seasonality and a higher level of annual precipitation, which increased the survivability of mosquitos in the winter, and the viability of mosquito larva, respectively. The two features taken together could exert an overwhelming effect on the ecology of mosquitoes, which was consistent with the findings of the high diversity of mosquito species in Southern China, where a semi-tropical city with warm, wet winters and tropical summers is predominant.

Precipitation is one of the most important elements for the breeding and development of mosquitoes, especially heavy late spring or summer rains, but not heavy winter rains. Further discretion for precipitation between seasons might help to attain a more accurate risk assessment. While grasslands, paddy fields, and forests constitute a natural habitat for mosquitoes, our analyses indicate a nontrivial role of human settlements. For example, densely populated areas are associated with elevated exposure to both *Cx. tritaeniorhynchus* and *Ae. albopictus*; in contrast, *An. minimus* and *Cx. pseudovishnui* prefer grasslands and forest with a low population density (Figure 1).

*Cx. tritaeniorhynchus* is by far the most widely distributed mosquito species, exposing over 80% of the nation’s population of the nation, except for the northwest (Appendix A). An enormous public health implication can be assumed, since *Cx. tritaeniorhynchus* harbors 19 arboviruses and acts as a competent vector for JEV as well, for which an infection case fatality ratio of 10–60% was achieved [61]. *Culex tritaeniorhynchus* feeding infectious blood meal had higher dissemination rates than other mosquito species for arboviruses, e.g., TMUV, WNV, and JEV, which means that *Culex tritaeniorhynchus* had a higher vector competence than other mosquitoes [62,63,64]. The vector competence of mosquitoes is associated with multiple endogenous features, including tissue barriers, the composition of the natural microbiome, and host preference [65,66,67]. Although previously confined to Asia, *Cx. tritaeniorhynchus* was suggested to be imported to Europe, such as to Greece [68]. *Ae. albopictus*, also known as Asian tiger mosquito, acted as the competent vector for DENV, CHIKV, YFV, and ZIKV, which have caused massive epidemics in history [69]. *Ae. albopictus* is also considered to be the most invasive mosquito species in the world, which although native to Asia, was additionally found in Europe and the Americas, such as Greece and the US [68,70]. Compared with *Ae. aegypti*, *Ae. albopictus* is more widely distributed in China, imposing imminent threats to temperate and subtropical regions (Appendix A).

Our findings on ecological drivers for JE and dengue, two notifiable arboviruses diseases in China, are consistent with previous ecological studies [71,72]. For example, grassland, annual mean temperature, and the presence probability of *Cx. tritaeniorhynchus* were drivers for JE [71], while annual mean temperature, annual cumulative precipitation, and the presence probability of *Ae. Albopictus* were drivers for dengue [72]. However, the current study adjusted the ecological models for the presence probability of competent vectors, which helps us to assess the net effects of socio-environmental factors on the ecology of the pathogens, controlling for their effects on the ecology of the competent vectors. The substantial contributions of the presence probability of mosquitos to the ecology of the two viruses clearly confirm the need for mosquito control to reduce the disease burden (Table 2). Disease response to climate variability varied with geographic area, and different climate variables appeared to play different roles in the disease transmission cycles. Therefore, finer scales of data might assist the specific surveillance and control for mosquitoes and MBDs. These findings demonstrate the need for an enhanced epidemiological understanding of multiple MBDs to inform disease prevention. An improved understanding of the relationship between environmental variability, mosquito density, and MBD transmission may assist public health decision-makers to plan and implement disease control interventions, and to allocate public health resources more effectively and efficiently.

In addition to JEV and DENV, other rare mosquito-borne arboviruses are threatening public health in China. For example, WNV was not only recorded in native probable cases, but also detected in mosquitoes and host animals in many provinces (Figure 3A and Appendix A). In addition, CHIKV, ZIKV, and YFV were found in imported cases, mosquitoes, and host animals (Figure 3A,B and Appendix A). Moreover, due to the synergy of the spatial expansion of competent vectors, the adaptive evolution of arboviruses, frequent international travels, and improved detection/diagnosis technology, more mosquito-borne arboviruses are likely to emerge in China in the future [5].

Our study is subject to limitations. Firstly, the survey locations of mosquitoes were extracted from literature and monographs, and thus unlikely to be representative of the entire country or ecological niches. However, the surveyed 1228 counties cover 43% of counties spread over all biogeographical zones in the nation (Appendix A), suggesting the adequate representativeness of the sampled sites. In addition, we weighted the counties by the estimated probabilities of being surveyed to reduce potential sampling bias in the BRT models. Secondly, the BRT models built an ensemble of the smaller decision tree to form a stronger classifier and improve prediction, but which inevitably caused potential over-fitting [73]. The solution of over-fitting is to adjust model configuration parameters by cross-validation optimization [73]. However, due to both larger sizes and the number of model runs, we cannot afford a full cross-validation optimization for all model configuration parameters. Thirdly, we adjusted GBRT models of mosquito-borne arboviruses for the presence probability of mosquitoes predicted by other ecological models, which does not necessarily reflect the actual density of mosquitoes. Furthermore, mosquito and arbovirus detection technologies likely differed across different studies, and recent studies might have employed more advanced technologies, creating potential bias in our ecological modeling results. Finally, while the AUC values are relatively high for all of the models that we fitted, AUC does not necessarily reflect the goodness of fit and could be misleading, as most surveys are cross-sectional, and the absence data are therefore associated with high uncertainty [74].

## 5. Conclusions

According to our study, it is necessary to expand the current field survey of mosquitoes to wider geographical coverage, and ideally, with longitudinal follow-up surveys, especially where the predicted possibility of mosquito species is high, but mosquito species investigations were missing. More resources should be allocated to detecting and controlling mosquitos harboring arboviruses implicative of high disease burden and imminent public health threats. Meanwhile, we recommend strengthening surveillance for MBDs by improving diagnostic technology and increasing public awareness in areas where model-predicted risk levels are high. Lastly, the importation of mosquito-borne arboviruses such as DENV, CHIKV, YFV, and ZIKV should be closely monitored, and their potential endemicity should be frequently examined in the high-risk areas of competent vectors, particularly southern cities with international ports and high population densities.

## Figures and Tables

**Figure 1 viruses-14-00691-f001:**
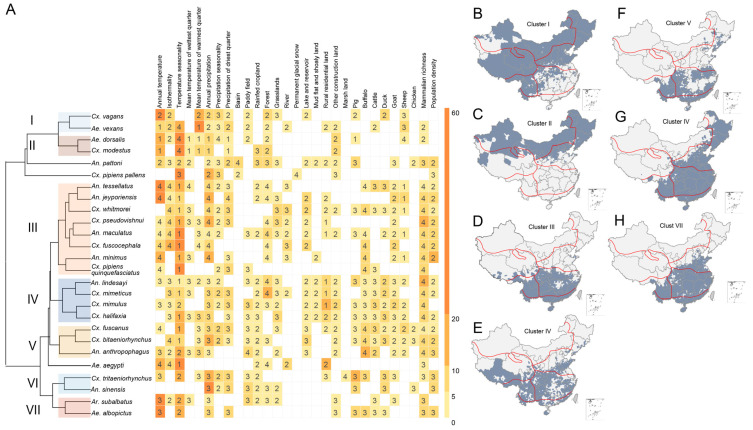
Clustering of mosquito species based on their ecological features and spatial distributions at the county level. Panels (**B**–**H**) indicate the spatial distributions of the seven clusters (clusters I–VII). The boundaries of the seven biogeographic zones are shown as red solid lines. The dendrogram in panel (**A**) displays the clusters I–VII of mosquito species. The features used for clustering are three quantities associated with each predictor in the BRT models. Two of the three quantities were displayed in panel (**A**) to indicate the possible level of ecological suitability: relative contributions (colors in ascending order from yellow to orange) and standardized median value of the predictor (numbers in the heatmap) among counties with mosquito occurrence (numbers 1–4 indicate the position of this median in reference to the quartiles of this predictor among all counties).

**Figure 2 viruses-14-00691-f002:**
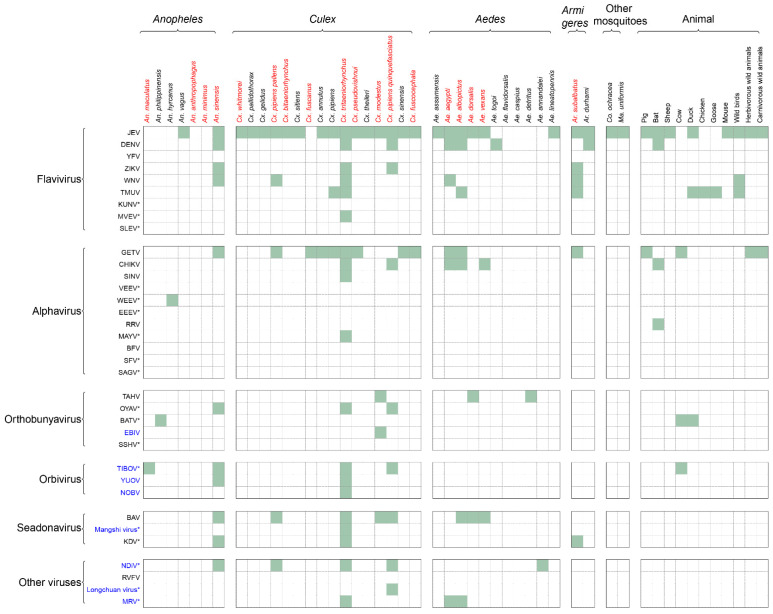
Mosquito species and animals for mosquito-borne viruses in China from 1954 to 2020. The virus names colored in blue indicate newly identified pathogens in the past two decades in China. “*” indicates arboviruses that have never been reported in human cases in China, but with the detection of viruses in mosquitoes or animals, or the detection of IgG in people. The mosquito names colored in red indicate species included in ecological models.

**Figure 3 viruses-14-00691-f003:**
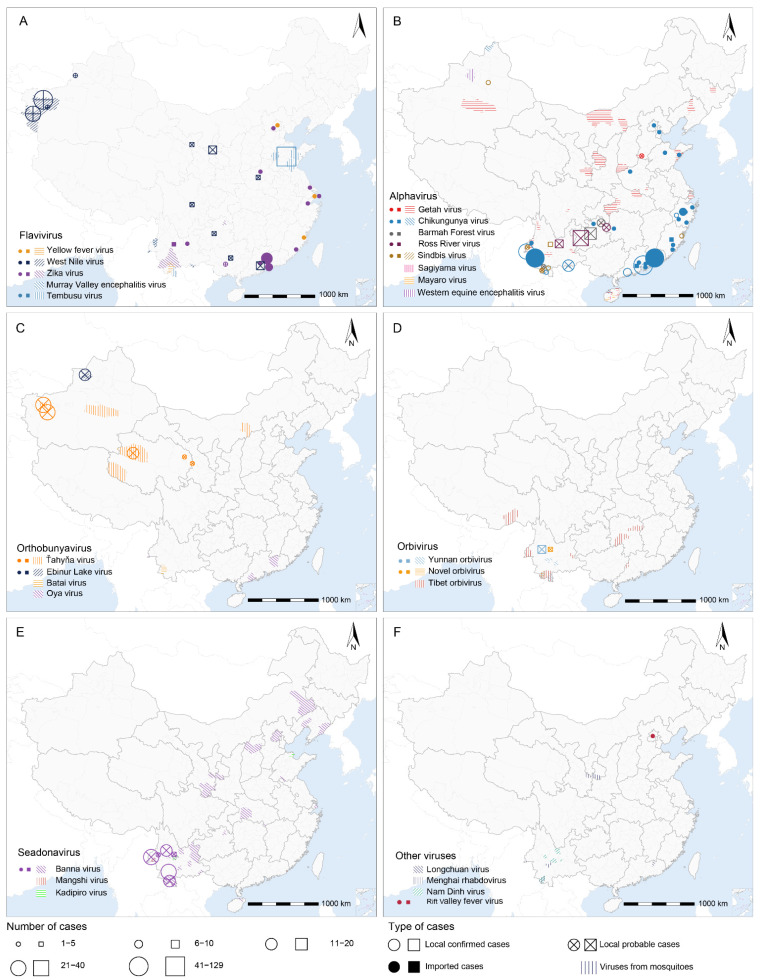
Distributions of reported human cases (circles or squares) and viral detections from mosquitos (shaded areas) of (**A**) *Flavivirus*; (**B**) *Alphavirus*; (**C**) *Orthobunyavirus*; (**D**) *Orbivirus*; (**E**) *Seadonavirus* and (**F**) other viruses in China in 1954–2020. Local confirmed (hollow circles or squares), local probable (hollow circles or squares with a cross), and imported (solid circles or squares) human cases are positioned at the center of prefectures/counties (circles) or provinces (square), depending on the finest available resolution. Human cases of JEV and DENV are not shown, as they are described in other figures. Source data are provided in Appendix A.

**Figure 4 viruses-14-00691-f004:**
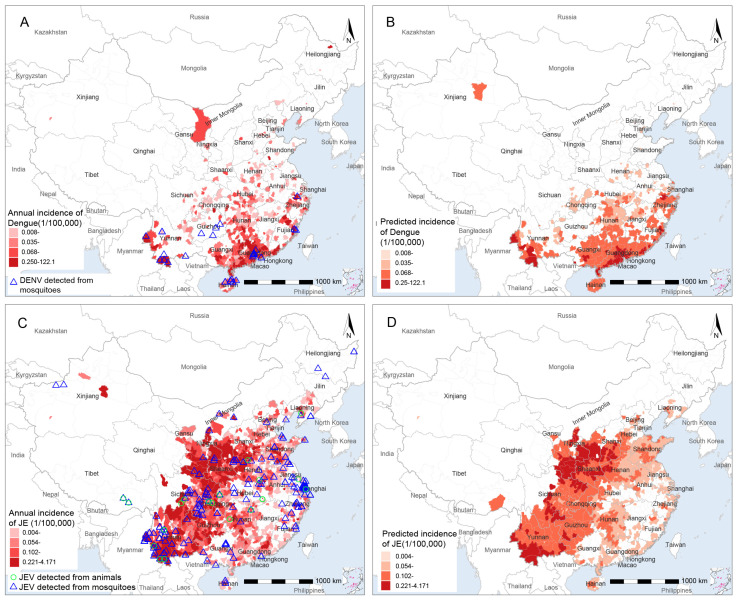
The reported and model-predicted distributions of dengue and JE at the county level in China. (**A**) Reported annual incidence rate of human dengue and locations of DENV detected from mosquitoes; (**B**) spatial distribution of model-predicted incidence rate of dengue; (**C**) reported annual incidence rate of human JE and locations of JEV detected from mosquitoes and host animals; (**D**) spatial distribution of model-predicted incidence rate of JE.

**Table 1 viruses-14-00691-t001:** The average testing areas under curve (AUC) of the BRT models and predicted numbers, coverage areas and population sizes of affected counties for the 26 most prevalent mosquito species in China.

Mosquito Species	Average AUC (2.5–97.5% Percentiles)	Predicted/Observed (Relative Difference %)		
		Number of Counties	Coverage Area (10 000 km ^2^)	Population Size (million)
*Armigeres subalbatus* ^c^	0.84 (0.79, 0.88)	1324/373 (255.0)	226.2/80.7 (180.3)	739.8/216.3 (242.0)
*Anopheles sinensis* ^a,b,c^	0.87 (0.81, 0.92)	1930/751 (157.0)	361.5/182.5 (98.1)	978.1/430.1 (127.4)
*An. anthropophagus*	0.90 (0.87, 0.94)	469/248 (89.1)	81.8/51.2 (59.8)	256.2/150.8 (69.9)
*An. minimus*	0.86 (0.82, 0.89)	631/236 (167.4)	137.1/55.7 (146.1)	290.1/131.7 (120.3)
*An. maculatus*	0.88 (0.82, 0.93)	282/124 (127.4)	82.0/36.7 (123.4)	108.2/51.5 (110.1)
*An. pattoni*	0.87 (0.82, 0.93)	411/117 (251.3)	83.1/30.7 (170.7)	181.3/63.7 (184.6)
*An. lindesayi*	0.79 (0.72, 0.86)	280/105 (166.7)	75.2/31.7 (137.2)	104.5/45.6 (129.2)
*An. jeyporiensis*	0.91 (0.85, 0.95)	242/104 (132.7)	54.2/24.2 (124.0)	120.1/57.6 (108.5)
*An. tessellatus*	0.91 (0.87, 0.95)	236/97 (143.3)	52.4/22.3 (135.0)	115.4/39.2 (194.4)
*Culex tritaeniorhynchus* ^a,c^	0.83 (0.77, 0.87)	1924/587 (227.8)	305.1/131.3 (132.4)	1042.8/330.6 (215.4)
*Cx. pipiens quinquefasciatus*	0.90 (0.85, 0.93)	930/384 (142.2)	193.5/87.0 (122.4)	465.5/206.2 (125.8)
*Cx. pipiens pallens* ^a,c^	0.94 (0.91, 0.97)	1695/381 (344.9)	332.7/111.8 (197.6)	856.9/234.0 (266.2)
*Cx. bitaeniorhynchus*	0.68 (0.63, 0.73)	560/250 (124.0)	114.8/57.3 (100.3)	275.2/136.2 (102.1)
*Cx. vagans* ^a,b^	0.77 (0.69, 0.84)	1397/186 (651.1)	582.6/78.0 (646.9)	546.9/92.9 (488.7)
*Cx. halifaxia*	0.76 (0.70, 0.82)	590/161 (266.5)	123.8/40.1 (208.7)	285.1/78.1 (265.0)
*Cx. modestus* ^b^	0.93 (0.88, 0.96)	561/133 (321.8)	336.2/118.5 (183.7)	184.8/57.1 (223.6)
*Cx. fuscanus*	0.80 (0.74, 0.86)	429/129 (232.6)	85.0/29.0 (193.1)	233.7/67.5 (246.2)
*Cx. mimeticus*	0.75 (0.65, 0.84)	402/123 (226.8)	113.0/35.4 (219.2)	151.4/54.4 (178.3)
*Cx. pseudovishnui*	0.82 (0.75, 0.88)	240/108 (122.2)	65.8/31.1 (111.6)	77.9/40.4 (92.8)
*Cx. fuscocephala*	0.85 (0.75, 0.92)	293/99 (196.0)	72.8/33.2 (119.3)	133.5/38.5 (246.8)
*Cx. whitmorei*	0.77 (0.70, 0.84)	261/95 (174.7)	59.5/24.8 (139.9)	106.5/42.0 (153.6)
*Cx. mimulus*	0.76 (0.68, 0.82)	459/82 (459.8)	158.1/20.7 (663.8)	200.0/37.3 (436.2)
*Aedes albopictus* ^a,c^	0.87 (0.83, 0.90)	1374/555 (147.6)	197.0/102.7 (91.8)	796.9/343.5 (132.0)
*Ae. vexans* ^b^	0.82 (0.76, 0.87)	953/247 (285.8)	436.5/140.8 (210.0)	329.7/107.9 (205.6)
*Ae. dorsalis* ^b^	0.93 (0.89, 0.96)	624/111 (462.2)	390.7/120.7 (223.7)	211.4/47.3 (346.9)
*Ae. aegypti*	0.95 (0.84, 1.00)	103/30 (243.3)	23.1/7.8 (196.2)	62.3/15.1 (312.6)

^a^ Top 5 mosquito species affecting the greatest numbers of counties. ^b^ Top 5 mosquito species affecting the widest areas. ^c^ Top 5 mosquito species affecting the largest population sizes. The predicted numbers are compared with the actual observations from field surveys and the relative differences (%) are given in parentheses.

**Table 2 viruses-14-00691-t002:** Mean (standard deviation) relative contributions of major factors to the spatial distributions of dengue and JE, estimated by two-stage GBRT models.

Category	Variable	Dengue (Relative Contributions %) #		Japanese Encephalitis (Relative Contributions %) #	
		Stage 1	Stage 2	Stage 1	Stage 2
Environmental	Basin (binary variable)			3.27 (2.20)	3.65 (1.55)
	Paddy field (%)			2.27 (0.48)	
	Rainfed cropland (%)	3.05 (0.59)		3.20 (0.84)	
	Forest (%)		1.82 (0.58)	3.20 (0.62)	
	Grasslands (%)				23.99 (8.87)
	River (%)			2.52 (0.51)	
	Rural residential land (%)				3.69 (0.96)
	Other construction land (%)				2.29 (0.46)
Ecoclimatic	Annual mean temperature	16.25 (4.88)	14.92 (6.65)	17.17 (6.52)	2.68 (0.54)
	Isothermality	3.43 (0.68)	3.82 (1.33)		2.54 (0.44)
	Temperature seasonality		3.00 (2.23)	8.78 (2.17)	
	Mean temperature of wettest quarter		11.29 (2.62)		2.05 (0.51)
	Mean temperature of warmest quarter	34.14 (7.35)			6.34 (4.46)
	Annual cumulative precipitation	7.00 (5.60)	27.30 (8.36)		
	Precipitation seasonality	2.71 (0.62)	2.77 (1.41)	4.00 (0.65)	4.03 (1.33)
	Precipitation of driest quarter	4.62 (1.26)		9.84 (2.77)	5.16 (1.07)
Social	Index of case importation	-	5.85 (3.93)	-	-
	Proportion of women	2.18 (0.43)		2.75 (0.46)	
	Proportion of ≥60 years old	3.01 (0.62)			2.68 (0.62)
	Number of general hospitals	5.35 (1.30)			3.75 (0.64)
	Number of clinics	3.18 (0.70)			
Biological	Density of population	5.01 (1.06)	4.03 (0.84)	2.67 (0.62)	2.33 (0.55)
	Mammalian richness	2.93 (0.76)		6.86 (1.19)	4.46 (0.81)
	Density of pig			3.79 (0.79)	5.92 (2.29)
	Density of cattle		2.66 (1.02)		3.04 (0.50)
	Density of duck			3.40 (0.72)	5.09 (1.57)
	Density of goat		2.69 (1.73)	3.10 (0.63)	
	Density of sheep		2.60 (1.77)	2.37 (0.52)	
	Density of chicken				16.29 (8.28)
	Presence of *Cx. tritaeniorhynchus* ^&^	-	-	16.11 (7.08)	
	Presence of *An. sinensis* ^&^	-	-	2.61 (0.53)	
	Presence of *Cx. pipiens quinquefasciatus* ^&^	-	-	2.08 (0.61)	
	Presence of *Ae. albopictus* ^&^	7.13 (1.15)	17.27 (7.56)	-	-

# The relative contributions were indicated as mean (standard deviation). “-” factors were not included in the whole model. ^&^ The presence of predominant mosquito species indicated the occurrence probability of each species predicted by the model. Stage 1 models the presence/absence of any reported human case, and stage 2 models the annual average incidences from 2014 to 2018 among presence locations. Mammalian richness indicated the number of mammal species.

## Data Availability

The data presented in this study are available in Appendix A.

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
