# Peer review of "Mapping the Distributions of Mosquitoes and Mosquito-Borne Arboviruses in China"

_viruses, 2022, doi:10.3390/v14040691_

Round 1
Reviewer 1 Report
In this manuscript, Wang et al examine the occurrence locations and times of mosquitoes and mosquito-borne viruses in mainland China over a 60+ year period from 1954. After mapping the mosquito occurrence at a county level, the authors then used machine learning algorithms to assess contributions of ecoclimatic and socioenvironmental factors to the spatial distributions of 26 predominant mosquito species and two MBDs associated with high disease burden.
Overall, this was an interesting paper and a well-organized manuscript. I have no serious concerns with the methodologies or any major issues with the manuscript in general.
However, some minor points need to be addressed:
In section 2.3, can the authors elaborate on why they call this model a “case-control” model? With the explanation/description provided, wouldn’t describing it as a classification model be more appropriate? Just designating something as ‘case’ and ‘control’ does not adequately explain why this model was described as such.
While considered at some length in the discussion, the authors can perhaps also elaborate on the limitations of the boosted tree model (eg. over-fitting, or the fact that cross validation isn’t possible in this instance.)
Other spelling/ typos-
Line 247: “Getah” instead of “getah”
Line 358: “assemble” instead of “assembled”
Line 359: “humans” instead of “human”
Author Response
Response to Reviewer 1 Comments
Point 1: In this manuscript, Wang et al examine the occurrence locations and times of mosquitoes and mosquito-borne viruses in mainland China over a 60+ year period from 1954. After mapping the mosquito occurrence at a county level, the authors then used machine learning algorithms to assess contributions of ecoclimatic and socioenvironmental factors to the spatial distributions of 26 predominant mosquito species and two MBDs associated with high disease burden.
Overall, this was an interesting paper and a well-organized manuscript. I have no serious concerns with the methodologies or any major issues with the manuscript in general.
However, some minor points need to be addressed:
In section 2.3, can the authors elaborate on why they call this model a “case-control” model? With the explanation/description provided, wouldn’t describing it as a classification model be more appropriate? Just designating something as ‘case’ and ‘control’ does not adequately explain why this model was described as such.
Response: We appreciate the reviewer’s positive comments and helpful suggestions. As suggested by the reviewer, we have supplemented the detailed information on the Boosted Regression Tree model as a classification model in the revised manuscript (Page 4, Lines 199–204).
Point 2: While considered at some length in the discussion, the authors can perhaps also elaborate on the limitations of the boosted tree model (eg. over-fitting, or the fact that cross validation isn’t possible in this instance.)
Response: Many thanks for your helpful suggestions. We have supplemented the details about limitations of the Boosted Regression Trees model in Discussion section (Page 19, Lines 733–738)
Point 3: Other spelling/ typos-
Line 247: “Getah” instead of “getah”
[Response] Many thanks for the reviewer’s correction. Done as suggestion (Page 11, Line 452).
Point 4: Line 358: “assemble” instead of “assembled”
[Response] Many thanks for the reviewer’s correction. Done as suggestion (Page 18, Line 649).
Point 5:Line 359: “humans” instead of “human”
[Response] Many thanks for the reviewer’s correction. Done as suggestion (Page 18, Line 650).

Reviewer 2 Report
General advice
I am truly amazed by the work providing by the team here. There are an amount of data and of shared data that are very very consequent.
Unfortunately, I am recommending the authors to resubmit a new version, longer, more accurate of their work. There are 133 pages of Supplementary data with 944 references, which are almost all about the presence of the mosquito species. It should worth to publish only this titanic and atoning work.
Moreover, the first part of the Supplementary Data (Supplementary Materials and Methods AND Supplementary Results – 9 pages in total) should be incorporated in a way to the manuscript. There are really useful informations for the comprehension of the article.
The only scientific critic (except minor ones following) is to understand how the study is based on so few human cases? Is that useful in the article to mention it as a result? How the viruses were detected, and where : on mosquitoes, human, birds : all that should be described in the manuscript.
Another short/quick remark : it is also quite surprising to not find any mention to the Tumbusu viruses, transmit mainly by some Culex species (such as Culex gelidus present I,n China see references 125, 128, 159, 287, 356, 381, 439, 440, 528, 548, 549, 579, 593, 621, 645, 648, 791, 822-825, 835, 894, 902, 926 in the Supplementary data).
Again, congrats to the authors, but a complete editorial work should be initiated.
Follow some minor comments until the end of the Mat & Met section
Abstract
At the reading of the abstract, there are already a lot of minor comments, and a general worry about the English writing of the manuscript.
Line 17: mosquito-borne viruses should be changed as ‘arbovirus’ within all the text
Line 21: is there only 339 mosquito species in China ?
Line 21: How only 34 arboviruses were defined?
Line 22: What is the scientific meaning of 18 species of mosquito-borne viruses ?
Line 22: It is quite strange to find Culex tritaeniorhynchus more than Culex pipiens/quinnquefasciatus, or even Aedes aegypti or Aedes albopictus. What are the main reasons ? It should be explain somewhere as it already apeear of a limit of the methodology used.
Lines 25-27: “The model-predicted suitable 25 habitats are 60‒664% larger in size than what have been observed, indicating the possibility of severe 26 under-detection.” Based on what : How the model was validated ?
Lines 27-29: “The spatial distribution of major mosquito species in China is likely under-estimated by current field observations. More active surveillance is needed to closely monitor competent mosquito vectors of MBVs to mitigate the risk of local endemicity of MBDs.” Always. It is impossible to obtain a spatial distribution of mosquito species representative even to the diseases distribution, even with an important catching effort. The spatial representability of the trapping areas is so small. It is not possible. That’s why we are developing model. The main important work should be on specific areas, where the model define the presence of specific species: there, we should catch to validate/unvalidate the models.
- Introduction
Lines 34-39: The first sentence is already 5 lines long. It has to be cut in at least 2 sentences and rewritten. …96 million dengue, 693,000 chikungunya fever, 500,000 Zika, 200,000 yellow fever, 68,000 Japanese encephalitis and 38 million cases per year [1-4].
Line 39: mosquito-borne viruses (MBVs) : why using this term ? The term arbovirus is generally used and totally accepted by the scientific community
Line 39-40: The assertion “Nearly all mosquito-borne viruses (MBVs) are RNA viruses” is a little naïve, and could not be expressed in that way. Insect specific virus are certainly also within the group MBV. Are we sure today about this sentence ?
Moreover, this sentence in its globality cannot be explained by only one example in the following sentence. Moreover, it does not express the idea. It express one example of a mutation, not a higher mutation rate…
Lines 45-48: I can not understand the meaning of the sentences and the idea supporting by the author. Please rephrase.
The introduction is too short : there are no information on the most important mosquito borne diseases in China, on the mosquito studies, number and genus present in China. These are classical knowledge important. We only have data on China. There is no information about the surrounding countries or even data in Asia for the diseases.
Moreover, these work has already be done in some other countries, it could be great to develop the introduction with knowledge coming from other countries.
Only from my point of view, the introduction should be totally rewritten.
- Materials and Methods
2.1. Data sources
My first main concern is about the collection of the data. How could be assured of the quality of the data? What is the data tracing?
Moreover, I went to the following sites: http://www.phsciencedata.cn , http://cdc.nmic.cn/home.do , http://www.geodata.cn , and I cannot find any data on these websites. Moreover, at the following address http://www.fao.org and https://sedac.ciesin.columbia.edu, I didn’t find any data. All the data should be provided or group somewhere. All these data should be clearly accessible. It is totally impossible to control check.
Regarding the supplementary file, I am very worry by the second file. In the supplementary files 2, there are in total in China 10 confirmed cases, 37 confirmed imported cases and 374 local probable patients from 1954 to March 2021 (=421 cases in 67 years (around 6 cases per year) in a country of around 1.4 billion inhabitants). It is hard to believe. Finally, there are more described viruses (236 lines) than confirmed cases (47).
Also, in the supplementary file, on the 346 mosquito species taking into considerations in the 2858 different counties, 9540 presence were mentioned on 265916 data. Main of the data are 0 (absence of presence) and No data: How were treated this data regarding the effort of sampling, the absence of sampling etc… ?
2.3. Ecological modeling
Line 101: “For each of the 26 major mosquito species,” How were selected the 26 mosquito species?
Lines 103-104: “At the first step, we excluded records that 103 were detected only in prefectures or provinces.” I do not really understand here the meaning of this sentence. Don’t all mosquitoes are caught within province or districts? What is the impact of such exclusion?
Line 106: what is the purpose of this classification: “complete investigation and incomplete investigation”?
---
Author Response
Response to Reviewer 2 Comments
Point 1: General advice
I am truly amazed by the work providing by the team here. There are an amount of data and of shared data that are very very consequent.
Unfortunately, I am recommending the authors to resubmit a new version, longer, more accurate of their work. There are 133 pages of Supplementary data with 944 references, which are almost all about the presence of the mosquito species. It should worth to publish only this titanic and atoning work.
Moreover, the first part of the Supplementary Data (Supplementary Materials and Methods AND Supplementary Results – 9 pages in total) should be incorporated in a way to the manuscript. There are really useful informations for the comprehension of the article.
Response: Many thanks for the reviewer’s positive comments and helpful suggestion. We have moved Supplementary Materials and Methods and Supplementary Results from Supplementary Appendix to appropriate sections in the revised main text.
Point 2: The only scientific critic (except minor ones following) is to understand how the study is based on so few human cases? Is that useful in the article to mention it as a result? How the viruses were detected, and where: on mosquitoes, human, birds: all that should be described in the manuscript.
Response: Many thanks for the reviewer’s helpful suggestions. In China, there are 40 kinds of notifiable infectious diseases, including dengue fever and Japanese encephalitis, which had high disease burden. Other mosquito-borne diseases caused by arboviruses are generally neglected and underestimated because they are not included in the national surveillance system of notifiable infectious diseases. In order to determining possible risk areas for arboviruses, we have supplemented the information about detection of arboviruses in animals as well as serosurveys of people in the revised Result section (Pages 14–15, Lines 544–597), and the detection methods and locations of arboviruses detected from mosquitoes, humans and animals were also supplemented in the revised manuscript (Lines 136–142, 461–597). In addition, we have supplemented the description of distribution of genotypes for JEV and serotypes for DENV from humans, animals and mosquitoes in the revised Results section (Page 12, Lines 463–489).
Point 3: Another short/quick remark: it is also quite surprising to not find any mention to the Tumbusu viruses, transmit mainly by some Culex species (such as Culex gelidus present in China see references 125, 128, 159, 287, 356, 381, 439, 440, 528, 548, 549, 579, 593, 621, 645, 648, 791, 822-825, 835, 894, 902, 926 in the Supplementary data).
Response: Many thanks for reviewer’s valuable comments. We apologize for the omission of Tembusu virus in the previous version. We have included Tembusu virus into the revised manuscript. In total, 1604 literature was searched by terms of “Tembusu virus” and “China”, including 1383 in Chinese and 221 in English, and 147 literatures and 377 records was supplemented to our updated database. In addition, we have supplemented the details about Tembusu virus in Results section (Lines 497–498, 551–553, 585–588) and updated the figures accordingly (Figures 2, 3 and Figures S43, S44).
Point 4: Again, congrats to the authors, but a complete editorial work should be initiated.
Response: We appreciate the reviewer’s encouragement and suggestion. We have conducted a complete editorial work for our manuscript.
Point 5: Follow some minor comments until the end of the Mat & Met section
Abstract
At the reading of the abstract, there are already a lot of minor comments, and a general worry about the English writing of the manuscript.
Line 17: mosquito-borne viruses should be changed as ‘arbovirus’ within all the text
Response: Many thanks for the reviewer’s helpful suggestion. We had corrected the name throughout the manuscript. We have also used “mosquito-borne arbovirus” only when a possible misunderstanding of “arbovirus” as viruses transmitted by other arthropods such as ticks exists.
Point 6: Line 21: is there only 339 mosquito species in China?
Response: Many thanks for the reviewer’s comment. According to “China Species Library”, the first authoritative database of all Chinese biological species, there are 377 mosquito species of 18 genus in China (https://species.sciencereading.cn/biology/v/searchResult/122/DW.html?sublibID=JSDWM). In our study, we aim to map the distribution of mosquito species at the county level, some mosquito species missing specific location at the county level were excluded in the current study. In addition, as the limitation on literature access, our study focused on the geographical scope of the mainland of China, some mosquitoes exclusively found in Taiwan, Hongkong and Macao were not included in our database. We have provided detailed information in the Methods section in the revised manuscript. (Page 3, Lines 128–131)
Point 7: Line 21: How only 34 arboviruses were defined?
Response: Many thanks for the reviewer’s valuable comment. Due to both enormous work time and volume, we only collected the information about mosquito-borne arboviruses that can infect humans. Other insect-specific viruses (e.g., Yichang virus) and arboviruses which transmit only to animals (e.g., Akabane virus, only infective for ruminant animals) were excluded in current study. We have supplemented the definition of 35 arboviruses in Materials and Methods section (Page 3, Lines 132-134).
Point 8: Line 22: What is the scientific meaning of 18 species of mosquito-borne viruses?
Response: Many thanks for the reviewer’s helpful comment. As we mentioned in Discussion section (Page 18, Lines 685–689), Culex tritaeniorhynchus presented an enormous underlying threat for public health, as it harbors 18 species of arboviruses and act as a competent vector for JEV as well, for which infection a case fatality ratio of 10‒60% was achieved. Moreover, Culex tritaeniorhynchus is by far the most widely distributed mosquito species, exposing over 80% of the nation’s population. Therefore, Culex tritaeniorhynchus should be closely monitored and strict control in areas where model-predicted risk levels are high.
Point 9: Line 22: It is quite strange to find Culex tritaeniorhynchus more than Culex pipiens/quinnquefasciatus, or even Aedes aegypti or Aedes albopictus. What are the main reasons? It should be explained somewhere as it already appear of a limit of the methodology used.
Response: Many thanks for the reviewer’s helpful suggestions. Based on comparison of vector competence in the laboratory, the probable reason that Culex tritaeniorhynchus harbored more arboviruses than other predominant mosquito species is the stronger vector competence than other mosquito species. Culex tritaeniorhynchus feeding infectious blood meal had higher dissemination rates than other mosquito species for arboviruses, e.g., Temsumu virus, West Nile virus and Japanese encephalitis virus. Vector competence of mosquitoes is associated with multiple endogenous features, including tissue barriers, composition of natural microbiome and host preference. We have discussed this issue in the revised manuscript (Pages 18–19, Lines 689–694) and six references (References 62–67).
Point 10: Lines 25-27: “The model-predicted suitable 25 habitats are 60‒664% larger in size than what have been observed, indicating the possibility of severe 26 under-detection.” Based on what: How the model was validated?
Response: Many thanks for the reviewer’s helpful comment. In the modeling step, a training set with 75% of data points was randomly selected by bootstrapping without replacement, and the remaining 25% served as a test set. The areas under the curve (AUC) and the partial AUC of the test sets were calculated to validate the effectiveness of model. We supplemented the validation of models in Materials and Methods section (Pages 5–6, Lines 249–251, 259–267).
Point 11: Lines 27-29: “The spatial distribution of major mosquito species in China is likely under-estimated by current field observations. More active surveillance is needed to closely monitor competent mosquito vectors of MBVs to mitigate the risk of local endemicity of MBDs.” Always. It is impossible to obtain a spatial distribution of mosquito species representative even to the disease distribution, even with an important catching effort. The spatial representability of the trapping areas is so small. It is not possible. That’s why we are developing model. The main important work should be on specific areas, where the model define the presence of specific species: there, we should catch to validate/unvalidate the models.
Response: Many thanks for reviewer’s helpful suggestion. We have modified the sentence as suggested in Abstract section (Page 1, Lines 28–30). In addition, we have also supplemented the idea in Conclusions section (Page 20, Lines 748–751).
Point 12: Introduction
Lines 34-39: The first sentence is already 5 lines long. It has to be cut in at least 2 sentences and rewritten. …96 million dengue, 693,000 chikungunya fever, 500,000 Zika, 200,000 yellow fever, 68,000 Japanese encephalitis and 38 million cases per year [1-4].
Response: Many thanks for reviewer’s helpful suggestion. Done as suggestion (Page1, Lines 35–40).
Point 13: Line 39: mosquito-borne viruses (MBVs): why using this term? The term arbovirus is generally used and totally accepted by the scientific community.
Response: Many thanks for the reviewer’s helpful suggestion. We had corrected the name throughout the manuscript. We have also used “mosquito-borne arbovirus” only when a possible misunderstanding of “arbovirus” as viruses transmitted by other arthropods such as ticks exists.
Point 14: Line 39-40: The assertion “Nearly all mosquito-borne viruses (MBVs) are RNA viruses” is a little naïve, and could not be expressed in that way. Insect specific virus are certainly also within the group MBV. Are we sure today about this sentence?
Response: Many thanks for the reviewer’s helpful comments. Actually this sentence is too ambiguous to correctly express idea that we want to say. We have modified this sentence as suggested in the Introduction section (Page 1, Lines 40–42).
Point 15: Moreover, this sentence in its globality cannot be explained by only one example in the following sentence. Moreover, it does not express the idea. It express one example of a mutation, not a higher mutation rate.
Response: Many thanks for the reviewer’s helpful suggestion. We have modified this sentence as suggested (Pages 1–2, Lines 42–49) and supplemented three references (Reference 6–8) in Introduction section.
Point 16: Lines 45-48: I can not understand the meaning of the sentences and the idea supporting by the author. Please rephrase.
Response: Many thanks for the reviewer’s helpful suggestion. I am sorry that this sentence is too ambiguous to correctly express idea that we want to say. We have modified this sentence in Introduction section (Page 2, Lines 68–71).
Point 17: The introduction is too short: there are no information on the most important mosquito borne diseases in China, on the mosquito studies, number and genus present in China. These are classical knowledge important. We only have data on China. There is no information about the surrounding countries or even data in Asia for the diseases.
Response: Many thanks for the reviewer’s helpful suggestion. Done as required (Page 2, Lines 50–66).
Point 18: Moreover, these work has already be done in some other countries, it could be great to develop the introduction with knowledge coming from other countries.
Response: Many thanks for the reviewer’s helpful suggestion. We have supplemented the studies about mosquito species investigation in other countries in Introduction section (Page 2, Lines 72–78).
Point 19: Only from my point of view, the introduction should be totally rewritten.
Response: Many thanks for reviewer’s helpful suggestion. Done as suggestion.
Point 20: Materials and Methods
2.1. Data sources
My first main concern is about the collection of the data. How could be assured of the quality of the data? What is the data tracing?
Response: Many thanks for reviewer’s helpful comments. For collection of data, each article was carefully reviewed by two team members independently to collect relative information. Any disagreement between the two staff members was resolved by discussion and consensus among the reviewers and other co-authors. Only studies with clearly identifiable results, i.e., presence or absence, time and location of mosquito species or MBVs, were included in our database. For articles containing ambiguous data, the original authors were contacted for clarification; if the ambiguity was not clarified, the data in question were excluded from our database. We conducted above process of collection to assure the quality of the data. We also have supplemented the details about process of data collection in Materials and Methods section (Page 3, Lines 113–131).
Point 21: Moreover, I went to the following sites: http://www.phsciencedata.cn , http://cdc.nmic.cn/home.do , http://www.geodata.cn , and I cannot find any data on these websites. Moreover, at the following address http://www.fao.org and https://sedac.ciesin.columbia.edu, I didn’t find any data. All the data should be provided or group somewhere. All these data should be clearly accessible. It is totally impossible to control check.
Response: Many thanks for the reviewer’s helpful comments. The website provided in Materials and Methods are the official website of the data source organizations. I am sorry that website of data is not directly accessible. We have deleted the fuzzy website and supplemented the direct websites of data in Table S2.
Point 22: Regarding the supplementary file, I am very worry by the second file. In the supplementary files 2, there are in total in China 10 confirmed cases, 37 confirmed imported cases and 374 local probable patients from 1954 to March 2021 (=421 cases in 67 years (around 6 cases per year) in a country of around 1.4 billion inhabitants). It is hard to believe. Finally, there are more described viruses (236 lines) than confirmed cases (47).
Response: Many thanks for the reviewer’s helpful suggestion. In fact, the number of cases is more than that the reviewer mentioned. There are in total in China 257 confirmed cases (including updated 63 cases of Tembusu virus), 253 confirmed imported cases and 374 local probable patients. We guess that the reviewer calculated the number of rows in confirmed cases and confirmed imported cases instead of the number of cases. In addition, diseases with highest incidence rate are dengue fever and Japanese encephalitis, which are included in the list of notifiable infectious diseases in China, but not included in the supplementary files. Moreover, compared with confirmed and probable cases, there were more people with positive detection in serological investigation. 18 of 35 arboviruses were not found in human cases but detected from animals, mosquitoes or serological investigation of people. We have supplemented the description of arboviruses from animals or serological investigation in Result section (Pages 14–15, Lines 544–597).
Point 23: Also, in the supplementary file, on the 346 mosquito species taking into considerations in the 2858 different counties, 9540 presence were mentioned on 265916 data. Main of the data are 0 (absence of presence) and No data: How were treated this data regarding the effort of sampling, the absence of sampling etc?
Response: Many thanks for reviewer’s helpful suggestion. The presence of specific mosquito species in given counties was defined as any evidence of occurrence in complete and incomplete investigations, and the absence of specific mosquito species was defined as no occurrence in complete investigations. Missing of specific mosquito species in given counties included two conditions, one where there were never any complete and incomplete investigation, another where incomplete investigation was conducted and specific mosquito species was not found. Counties with only incomplete investigations will not contain any absence of all mosquito species. In this way, bias for scope of mosquito species investigated was decreased between complete and incomplete investigation. However, the bias of sampling was inevitably presented in database. So based on this data, counties without sampling should be conducted active field investigation. In modeling of mosquito species, we conducted a logistic model to calculated the possibility of sampling which was used in weighted modeling. Therefore, the model-predicted presence and absence are complete for all counties and counterbalanced by weighted modeling.
Point 24: 2.3. Ecological modeling
Line 101: “For each of the 26 major mosquito species,” How were selected the 26 mosquito species?
Response: Many thanks for reviewer’s helpful suggestion. The selection of 25 predominant species was based on the more than 80 observed counties. Additionally, Aedes aegypti was selected for modeling as competent vector for dengue virus, zika virus and chikungunya virus even though Aedes aegypti is observed in only 30 counties. We have supplemented the selection strategy of mosquito species used to modeling in Materials and Methods section (Page 4, Lines 197–199).
Point 25: Lines 103-104: “At the first step, we excluded records that 103 were detected only in prefectures or provinces.” I do not really understand here the meaning of this sentence. Don’t all mosquitoes are caught within province or districts? What is the impact of such exclusion?
Response: Many thanks for reviewer’s helpful suggestion. The BRT models were conducted at county levels, and the records at the provinces or prefectures cannot be included in modeling. For example, one record of Cx. tritaeniorhynchus was reported in Shanxi Province, but we don’t know which county the mosquitoes were collected in, as the limitation of source literature, so this record was excluded from our study. We have moved this sentence from 2.3 section to 2.1 section (Page 3, Lines 128–129).
Point 26: Line 106: what is the purpose of this classification: “complete investigation and incomplete investigation”?
Response: Many thanks for reviewer’s helpful suggestion. This classification was used to define the presence and absence of mosquitoes in counties. The presence of specific mosquito species in given counties was defined as any evidence of occurrence in complete and incomplete investigations, and the absence of specific mosquito species was defined as no occurrence in complete investigations. Missing of specific mosquito species in given counties included two conditions, one where there were never any complete and incomplete investigation, another where incomplete investigation was conducted and specific mosquito species was not found. We have supplemented the example of Aedes albopictus to express this idea in Materials and Methods section (Page 5, Lines 213–218).
